# Aerobic Exercise Training with Brisk Walking Increases Intestinal Bacteroides in Healthy Elderly Women

**DOI:** 10.3390/nu11040868

**Published:** 2019-04-17

**Authors:** Emiko Morita, Hisayo Yokoyama, Daiki Imai, Ryosuke Takeda, Akemi Ota, Eriko Kawai, Takayoshi Hisada, Masanori Emoto, Yuta Suzuki, Kazunobu Okazaki

**Affiliations:** 1Department of Environmental Physiology for Exercise, Osaka City University Graduate School of Medicine, 3-3-138 Sugimoto, Sumiyoshi-ku, Osaka-shi, Osaka 558-8585, Japan; e-morita@pt-u.aino.ac.jp (E.M.); dimai@sports.osaka-cu.ac.jp (D.I.); kawai@respiratorycontrol.com (E.K.); suzuki@sports.osaka-cu.ac.jp (Y.S.); okazaki@sports.osaka-cu.ac.jp (K.O.); 2Department of Physical Therapy Faculty of Health Science, Aino University, 4-5-4 Higashiohda, Ibaraki-shi, Osaka 567-0012, Japan; 3Research Center for Urban Health and Sports, Osaka City University, 3-3-138 Sugimoto, Sumiyoshi-ku, Osaka-shi, Osaka 558-8585, Japan; wind-05@med.osaka-cu.ac.jp; 4Department of Health and Sports Science, Osaka Electro-communication University, 1130-70 Kiyotaki, Shijonawate-shi, Osaka 575-0063, Japan; ota@osakac.ac.jp; 5TechnoSuruga Laboratory Company, Ltd., 330 Nagasaki, Shimizu-ku, Shizuoka-shi, Shizuoka 424-0065, Japan; tsl-contact@tecsrg.co.jp; 6Metabolism, Endocrinology and Molecular Medicine, Osaka City University Graduate School of Medicine, 1-4-3, Asahi-Machi, Abeno-ku, Osaka-shi, Osaka 545-8586, Japan; memoto@med.osaka-cu.ac.jp

**Keywords:** intestinal microbiota, intestinal *Bacteroides*, cardiorespiratory fitness, trunk muscle training, aerobic exercise training, brisk walking

## Abstract

This study examined the effect of an exercise intervention on the composition of the intestinal microbiota in healthy elderly women. Thirty-two sedentary women that were aged 65 years and older participated in a 12-week, non-randomized comparative trial. The subjects were allocated to two groups receiving different exercise interventions, trunk muscle training (TM), or aerobic exercise training (AE). AE included brisk walking, i.e., at an intensity of ≥ 3 metabolic equivalents (METs). The composition of the intestinal microbiota in fecal samples was determined before and after the training period. We also assessed the daily physical activity using an accelerometer, trunk muscle strength by the modified Kraus–Weber (K-W) test, and cardiorespiratory fitness by a 6-min. walk test (6MWT). K-W test scores and distance achieved during the 6MWT (6MWD) improved in both groups. The relative abundance of intestinal *Bacteroides* only significantly increased in the AE group, particularly in subjects showing increases in the time spent in brisk walking. Overall, the increases in intestinal *Bacteroides* following the exercise intervention were associated with increases in 6MWD. In conclusion, aerobic exercise training that targets an increase of the time spent in brisk walking may increase intestinal *Bacteroides* in association with improved cardiorespiratory fitness in healthy elderly women.

## 1. Introduction

“All disease begins in the gut”., a quotation from the ancient Greek physician Hippocrates, highlights the potential roles of intestinal microbiota in various disease risks, which have recently attracted considerable attention from researchers. The presence of an imbalanced, low-diversity, intestinal microbiota is known as dysbiosis and it is associated with a variety of pathologies, including constipation [1], obesity [2], diabetes [3], colon cancer [4], coronary artery disease [5], inflammatory bowel disease [6], and depression [7]. Aging also strongly affects the composition of the intestinal microbiota. In general, the intestinal microbiota of the elderly show reduced species diversity [8]. In addition, intestinal *Bifidobacterium* and *Bacteroides*, which are known to be related to obesity, are also reduced [8], which potentially contributes to the high prevalence of obesity in the elderly population. Overall, the intestinal microbiota could be regarded as an indicator of host health.

Multiple factors, including host genetics [9], method of childbirth (i.e., by vaginal delivery or caesarian section) [10], age [8], nutrition [11], and antibiotic intake [8], have been suggested to affect the composition of the intestinal microbiota. Recent studies demonstrated the association between exercise training, i.e., a low-cost health strategy, and lower risks of colon cancer [12,13], a disease that is known to at least partly arise from imbalanced intestinal microbiota [4]. Therefore, exercise may also have potential for modifying the composition of the intestinal microbiota, although these studies did not directly examine the effect of exercise on intestinal microbiota. 

In fact, animal studies have demonstrated the changes in the composition of the intestinal microbiota by exercise training [14,15,16]. A number of cross-sectional human studies have confirmed the associations between physical activity or cardiorespiratory fitness and the composition of the intestinal microbiota [17,18,19]. For example, rugby players were found to have a greater diversity of intestinal microbiota and an enlarged abundance of *Akkermansia*—which is known to prevent diabetes—when compared to sedentary adults [17]. Other studies showed that cardiorespiratory fitness or physical activity level is associated with greater microbial diversity in healthy humans [18,19,20]. Furthermore, trained elite race walkers show increased relative abundance of *Bacteroides,* in combination with high fat diet [21]. However, these studies did not examine the effect of exercise alone on intestinal microbiota independent of the dietary habits that may have the greater impact on intestinal microbiota than exercise. Therefore, the potential impact of exercise interventions on human intestinal microbiota has not been fully clarified.

In the present study, we examined the effects of exercise interventions on intestinal microbiotic composition in healthy elderly women. We hypothesized that an improvement of cardiorespiratory fitness would be crucial to exercise-induced changes in the intestinal microbiota. We compared the effects of two exercise modalities on the intestinal microbiota: aerobic exercise, which specifically enhances cardiorespiratory fitness, and trunk muscle training as a control condition to verify this hypothesis.

## 2. Materials and Methods 

### 2.1. Subjects

Thirty-two healthy sedentary women that were aged 65 years and over were recruited from the residents of Osaka City, Japan, by an advertisement in a local magazine. The selected 32 subjects voluntarily opted for enrollment in either of the two exercise programs, aerobic exercise training (AE) or trunk muscle training (control condition; TM). Prior to the study, none of the subjects engaged in a regular exercise for more than 1 h per week. Health status and the use of medication were assessed by structured interview. Applicants presenting a history of ischemic heart disease, chronic heart failure, stroke, severe hypertension, diabetes, or neuropsychiatric disorder were excluded from the study. Applicants who were judged by a physician to be unable or ill-equipped to participate in the exercise program were also excluded. Consequently, none of the 32 subjects was excluded. The Institutional Review Board of Osaka City University Graduate School of Medicine approved the study protocol (approval no. 3501, approved on August 30, 2016). The authors also confirm that all of the ongoing and related trials for this intervention are registered in the University Hospital Medical Information Network Clinical Trials Registry (UMIN 000023930). Written informed consent was obtained from all of the participants after explanation of the study purpose. The study protocol also conformed to the ethical guidelines of the 1975 Declaration of Helsinki.

### 2.2. Study Design

The study design involved a 12-week non-randomized, comparative trial, in which the allocation of the participants to either of the two exercise groups, AE and TM, was based on their own preference. This study was conducted between the first recruitment of the participants on 12 September 2016 and the final follow-up of the participants on 24 January 2018. Before study enrollment, all of the applicants visited our research center at Osaka City University for baseline measurements, e.g., body composition, motor ability, and clinical laboratory analyses, as well as an assessment of daily physical activity levels, nutrient intake, and bowel habits. In addition, fecal samples were collected. All of the baseline assessments were conducted at least 1 week before the first training session. Finally, 18 and 14 applicants who met the inclusion criteria were enrolled in the AE group and the TM group, respectively, after which they were started on the selected 12-week exercise programs. The measurements during the baseline session were repeated at least one week after the final session of the exercise program.

### 2.3. Exercise Intervention

The subjects in the TM group received a 1-h group training weekly for 12 weeks, which aimed at strengthening the trunk muscles. All of the sessions were held at Sumiyoshi Sports Center, a gymnasium located in Osaka City, and supervised by a trained instructor. A training session comprised 5–10 min. of warm-up, followed by 45 min. of targeted resistance training of the trunk muscles and 5–10 min. of cool down exercises. Figure 1 shows examples of the trunk muscle training. The training was composed of several kinds of exercises, including arching–swaying, plank, pelvic rotation in the supine position, and diagonal lifting while standing on all fours. The contraction duration was set at 3 to 5 s, and each exercise was performed in two sets of 10 repetitions. The subjects were also instructed to work out at home daily. Adherence to group sessions, as well as to the home exercises, was recorded weekly by the instructor throughout the 12-week intervention period.

The subjects in the AE group were instructed to perform 60 min. of brisk walking at an intensity of ≥ 3 metabolic equivalents (METs) daily for 12 weeks. They wore a three-axis accelerometer (Mediwalk^®^ MT-KT02DZ, TERUMO, Tokyo, Japan [22,23]) throughout the intervention period, except while sleeping and bathing, to record their daily number of steps and time that is spent in brisk walking. The instructor shared the accelerometer data with the participants once a week and was encouraged them to increase the intensity and duration of their brisk walking regimen gradually as much as possible. The subjects were also instructed to keep good posture while walking. 

### 2.4. Analysis of Intestinal Microbiota

The fecal samples were collected in a container with guanidine thiocyanate as a preservative solution (TechnoSuruga Laboratory, Shizuoka, Japan) and refrigerated at 4 °C until transfer to the laboratory within seven days. We conformed to the protocol [24] for the representative extraction of DNA from bacterial populations in feces. Terminal restriction fragment length polymorphism (T-RFLP) analyses to determine the relative abundance of intestinal microbiota phylogenetic groups from each fecal sample were performed at the TechnoSuruga Laboratory (Shizuoka, Japan) [25,26]. T-RFLP analysis is one of the most well-established and reliable 16S ribosomal RNA-based methods, especially when considering its high throughput and reproducibility. Briefly, the fecal samples (approximately 4 mg each) were suspended in a 1200 μL solution containing 100 mM Tris-HCl (pH 9.0), 40 mM ethylenediaminetetraacetic acid, 4 M guanidine thiocyanate, and 0.001% bromothymol blue. A FastPrep 24 device homogenized the Fecal solids in the suspension (MP Biomedicals, Irvine, CA, USA) with zirconia beads being set at 5 m/s for 2 min. DNA was then extracted from a 200 µL suspension using magLEAD 12gC (Precision System Science; Chiba, Japan). MagDEA^®^ Dx SV (Precision System Science) was used as the reagent in automatic nucleic acid extraction. PCR was performed with a Takara Thermal Cycler Dice TP650 (Takara Bio, Shiga, Japan) in 20 µL of a reaction mixture containing 1× PCR buffer, with each deoxynucleotide triphosphate at a concentration of 200 µM, 1.5 mM MgCl_2_, each primer at a concentration of 0.2 µM, 10 ng of fecal DNA, and 0.2 U of HotStarTaq DNA polymerase (Qiagen, Hilden, Germany). 5′ FAM-labeled 516f (5’-TGC-CAGCAGCCGCGGTA-3’; *Escherichia coli* positions 516−532) and 1510r (5’-GGTTACCTTGTTACGA-CTT-3’; *E. coli* positions 1510−1492) were the primers used. The amplification program used was as follows: preheating at 95 °C for 15 min, 35 cycles of denaturation at 95 °C for 30 s, annealing at 50 °C for 30 s, extension at 72 °C for 90 s, and finally, terminal extension at 72 °C for 10 min. Electrophoresis and purified using a MultiScreen PCR µ96 Filter Plate verified amplified DNA (Millipore, Billerica, MA, USA). The purified 16S rDNA amplicons were treated with 10 U of FastDigest BseLI (Thermo Fisher Scientific, Waltham, MA, USA) for 10 min. An ABI PRISM 3130xl genetic analyzer (Thermo Fisher Scientific) was used to analyze the resultant DNA fragments, i.e., fluorescent-labeled terminal restriction fragments (T-RFs). GeneMapper software (Thermo Fisher Scientific) was used to determine the T-RF length and the peak area for each sample. T-RFs were divided into 29 operational taxonomic units (OTUs). The individual OTUs were quantified as the percentage of all OTUs combined based on the area under the curve (% AUC). The reference database, Human Fecal Microbiota T-RFLP profiling (http://www.tecsrg-lab.jp/t_rflp_hito_OTU.html), was used to putatively match the bacteria in each classification unit to the corresponding OTU. T-RFLP analyses enabled the classification of the sampled intestinal microbiota into the following 10 groups: *Bifidobacterium*, *Lactobacillales*, *Bacteroides*, *Prevotella*, *Clostridium* cluster IV, *Clostridium* subcluster XIVa, *Clostridium* cluster IX, *Clostridium* cluster XI, *Clostridium* cluster XVIII, and others. 

### 2.5. Anthropometrical Measurements

The body mass index (BMI) was calculated as body weight/(height)^2^, as expressed in kg/m^2^. Bioelectrical impedance analysis using a body composition analyzer estiated the percentages of fat and muscle mass of the trunk and lower extremities (Nippon Shooter Ltd., Physion MD, Tokyo, Japan).

### 2.6. Physiological Performance

Quadriceps muscle strength was assessed using a strain gage dynamometer (ST-200S, MUL-TECH, Tokyo, Japan). Each subject performed two attempts on each leg and the maximum value of these four trials was marked for later analysis. The modified Kraus–Weber (K-W) test was used to assess trunk muscle strength [27]. This simple exercise test was based on the K-W Minimum test that was developed by Drs. Hans Kraus and Sonja Weber in the 1950s [28] to assess the strength and endurance of the trunk muscles. The trunk muscle strength of each subject was rated based on the total scores (full marks = 40) of the test (Appendix A).

Four physical performance tests were conducted to evaluate motor ability and fitness: maximal step length (MSL), Timed Up and Go (TUG) test, single-leg standing, and the 6-min. walk test (6MWT). MSL was determined as the maximum possible stride per step of a subject. In the TUG test, we measured the time that is required for a subject to stand up from a chair, walk 3 m, turn, walk back to the chair, and sit down. In single-leg standing, we measured the maximum time that a subject could stand on one leg. In case a subject continued single-leg standing for over 120 s, the test was discontinued. All of the functional tests were conducted twice and the best scores were marked for analysis. Cardiorespiratory fitness was evaluated by the 6MWT according to the guidelines of the American Thoracic Society [29]. In short, he subjects were instructed to walk back and forth on a 25-m course as fast as possible for 6 min under the supervision of a medical doctor. They were permitted to stop and rest in case of fatigue. The investigator encouraged the subjects with routine phrases (e.g., “you are doing well” and “keep up the good work”) once per minute during the test. The total distance (in meters) walked after 6 min. (6MWD) was recorded and used as an indicator of cardiorespiratory fitness, since performance on the 6MWD strongly correlates with peak oxygen uptake [30,31]. 

### 2.7. Daily Physical Activity Level

The parameters reflecting the daily physical activity level of the participants included the number of steps and the time spent in brisk walking—i.e., at an intensity of three METs or more—was estimated using the same three-axis accelerometer as that used during training in the AE group. This device also automatically calculates ethe nergy expenditure (EE) from METs based on a widely-accepted formula (EE (kcal) = 1.05 × METs × time (h) × body weight (kg) [32]). All of the subjects were instructed to wear the accelerometer throughout the one-week measurement period, except while sleeping and bathing, and to continue with daily activities as usual. The assessments were conducted before and after the 12-week intervention. The data, which were automatically stored on the device, were subsequently transferred to a computer while using specialized software (HR Joint^®^ Smile Data Vision, TERUMO, Tokyo, Japan). The mean daily values of all parameters recorded during the one-week monitoring period were used for further analysis.

### 2.8. Laboratory Measurements

The blood samples were collected at 9 AM under standardized 12-h fasting conditions. Serum samples were stored at −80 °C until further analysis. The hexokinase UV method measured the plasma glucose levels, whereas serum insulin levels were determined by chemiluminescent enzyme immunoassay. Serum triglycerides, low-density lipoprotein cholesterol (LDL-C), and high-density lipoprotein cholesterol (HDL-C) were determined by enzymatic methods. The homeostasis model assessment of insulin resistance (HOMA-IR), which is an established surrogate index of insulin resistance [33], was also determined. The HOMA-IR was obtained from fasting plasma glucose (FPG) and serum insulin (FIRI) levels according to the original method by Matthews et al. [34] while using the following formula:HOMA-IR = FPG (mmol/L) × FIRI (µU/mL)/22.5(1)
A higher HOMA-IR value represents higher insulin resistance.

### 2.9. Nutrient Intake

Nutrient intake was estimated using a food frequency questionnaire (FFQ), which the Japan Public Health Center-based Prospective Study developed and previously validated [35]. The FFQ consists of 138 food and beverage items and measures nine intake frequency categories: never or seldom, 1–3 times/month, 1–2 times/week, 3–4 times/week, 5–6 times/week, once/day, 2–3 times/day, 4–6 times/day, and more than seven times/day. All of the subjects were asked to complete the questionnaire before and after the 12-week intervention. FFQ data were analyzed with the help of Education Software Co., Ltd. (Tokyo, Japan), and then converted to quantitative estimates of the daily consumed amounts of energy, protein, lipid, carbohydrates, saturated fat, and dietary fiber.

### 2.10. Defecation Assessment

Defecation patterns were assessed using the Japanese version of the Constipation Assessment Scale (CAS-J), which was modified from the original scale that was developed by McMillan et al. [36] to assess constipation in Japanese populations [37]. The CAS-J comprises eight questions, i.e., “The abdomen appears distended or swollen”, “The amount of flatus”, “The frequency of defecation” “The rectum appears to be filled with feces”, “Pain of the anus during defecation”, “The amount of feces”, “Ease of defecation”, and “Diarrhea or watery stools”. Each item includes a three-point rating scale: 0 (“no problem”), 1 (“some problem”), and 2 (“severe problem”). Thus, the maximum possible CAS-J score is 16, with higher scores indicating more severe cases of constipation.

### 2.11. Statistical Analyses

The data are presented as median and interquartile ranges. Changes in clinical parameters and relative abundances of specific classes of intestinal microbiota following intervention in each group were examined by the Wilcoxon Signed-rank test. The Spearman’s rank correlation coefficient test examined the relationships between the parameters and changes in the relative abundance of specific types of intestinal microbiota. Stepwise regression analysis was also performed to identify the factors that determined the change in the relative abundance of specific microbiota. Finally, the Mann–Whitney U-test was used to compare the changes in the relative abundance of specific types of intestinal microbiota between the exercise groups according to the increase in time spent in brisk walking. All of the statistical procedures were performed using SPSS statistical software (version 24.0, IBM, New York, NY, USA). *P* values less than 0.05 were considered to be statistically significant. 

## 3. Results

### 3.1. Clinical Characteristics of the Subjects

Figure 2 shows the procedural flowchart of the enrollment, measurement, intervention, and data analysis of this study. Two participants in the TM group and one in the AE group dropped out during the intervention period. A total of 12 participants in the TM group and 17 participants in the AE group completed the study. We could confirm that all of the subjects in the TM group participated in 90% or more of the sessions and that the mean adherence to the home exercise was 96.0%. The mean percentage of attendance at weekly meetings with the instructor was 97.1% in the AE group. Table 1 summarizes the clinical characteristics of both groups. The median age was 70 (65–77) years in the TM group and 70 (66–75) years in the AE group. 

### 3.2. Changes in Body Composition, Muscle Strength, Physical Performance, and Daily Physical Activity Following the Intervention

Table 2 shows the changes in body composition, muscle strength, physical performance, and daily physical activity following the intervention in both groups. 

The number of steps (*p* = 0.004) and the time spent in brisk walking (*p* = 0.003), as well as the exercise-induced EE (*p* = 0.003), were significantly increased following the intervention in the AE group only. Total EE was significantly increased in the AE group (*p* = 0.012), while it was decreased in the TM group (*p* = 0.049) following the intervention. The K-W test scores (TM group: *p* = 0.008; AE group: *p* < 0.001) and 6MWD (TM group: *p* = 0.028; AE group: *p* = 0.001) were equally improved following the intervention in both groups. No further significant changes were observed in other parameters of motor ability following the interventions in either group.

### 3.3. Changes in Laboratory Measurements Following the Intervention

Table 2 shows the changes in the laboratory measurements following the intervention in both groups. FPG and blood levels of triglycerides, LDL-C, HDL-C, and insulin as well as HOMA-IR remained unchanged after the intervention in both groups. 

### 3.4. Changes in Nutrient Intake and Defecation Pattern Following the Intervention

Table 3 shows the changes in nutrient intake and the CAS-J scores following the intervention. Significant differences in nutrient intake patterns as well as total energy intake were found neither at baseline nor after the interventions. Regarding the patterns of defecation, the CAS-J scores were significantly decreased in the AE group only (*p* = 0.036) following the interventions. For individual components of the CAS-J, the score on “Ease of defecation” was significantly decreased following the intervention in the AE group. On the other hand, the score in “The rectum appears to be filled with feces” was significantly improved following the intervention in the TM group only.

### 3.5. Composition of Intestinal Microbiota

Figure 3 shows the composition of the intestinal microbiota in both groups. Following the interventions, the relative abundance of *Bacteroides* was significantly increased, and that of *Clostridium* subcluster XIVa was only decreased in the AE group. The relative abundance of *Clostridium* cluster IX was only significantly increased in the TM group. After the interventions, the relative abundance of other microbiota groups remained unchanged in both of the groups.

### 3.6. Relationship between Changes in the Parameters and Change in the Relative Abundance of Intestinal Bacteroides after the Intervention

We examined the relationships between age, the relative abundance of intestinal *Bacteroides* before the intervention (pre-*Bacteroides*), or the changes in the parameters that were modulated by the exercise intervention and the change in the relative intestinal abundance of *Bacteroides* (Δ%*Bacteroides*). Pre-*Bacteroides* was negatively correlated with Δ%*Bacteroides* (*r* = −0.519, *p* = 0.004) when analyzing all of the subjects combined. A significant positive correlation was also found between the change in 6MWD (Δ6MWD; *r* = 0.431, *p* = 0.020) or that in time spent in brisk walking (ΔTime spent in brisk walking; *r* = 0.371, *p* = 0.047) and the Δ%*Bacteroides* following the intervention in all subjects (Figure 4). There were no significant correlations between changes in other parameters and Δ%*Bacteroides* among all the subjects combined (Table 4).

To identify the factors that contribute to Δ%*Bacteroides*, we performed stepwise multiple regression analysis, in which Δ%*Bacteroides* was included as the dependent variable and age, pre-*Bacteroides*, Δ6MWD, and ΔTime spent in brisk walking were included as the possible independent variables. In this analysis, Δ6MWD (β = 0.370, *p* = 0.034) and pre-*Bacteroides* (β = −0.356, *p* = 0.041) were found to be independent contributors (*R*^2^ = 0.317).

### 3.7. Effect of Increased Daily Physical Activity on Changes in the Relative Abundance of Intestinal Bacteroides Following the Intervention in the AE Group

Although the improvement of 6MWD was observed in each group, a significant increase in the relative abundance of intestinal *Bacteroides* was only found in the AE group. Therefore, we focused on the effect of the increased time spent in brisk walking on Δ%*Bacteroides* following the intervention in the AE group. The subjects in the AE group were divided into two groups according to whether they had increased their time spent in brisk walking by more or less than 20 min. following the intervention. As shown in Figure 5, Δ%*Bacteroides* in subjects who added > 20 min. of time spent in brisk walking was greater than that in the subjects who added ≤ 20 min. (9.7% (4.7%–14.2%), *n* = 10 vs. –3.5% (–4.2% – 2.4%), *n* = 7; *p* = 0.025).

## 4. Discussion

The aim of the present study was to investigate whether exercise intervention modifies the composition of intestinal microbiota in healthy elderly women. Our main findings were that a 12-week aerobic exercise program that consists of daily episodes of brisk walking increased the relative abundance of intestinal *Bacteroides,* while improving cardiorespiratory fitness without any changes to nutrient intake. Moreover, the increase relative abundance of intestinal *Bacteroides* was especially marked in subjects who increased the time spent in brisk walking by more than 20 min. We also found that aerobic exercise improved the pattern of defecation independently of Δ%*Bacteroides*. Meanwhile, the elderly subjects who engaged in the trunk muscle training showed neither a significant Δ%*Bacteroides* nor changes in the defecation pattern.

To date, the primary findings of animal studies suggested that the level of exercise may modulate the composition of the intestinal microbiota. In rodents, six days of wheel running exercise increased *Bifidobacterium* and *Lactobacillus*, which are widely recognized as health-promoting intestinal bacteria [14]. It was also reported that the exercise-induced changes of the intestinal microbiota in mice depend on the exercise modalities (voluntary wheel running or forced treadmill running) and that voluntary wheel running reduced *Turicibacter* spp., which are associated with immune dysfunction and bowel diseases [15]. Another study demonstrated that a six-week schedule of interval treadmill running in mice enhanced the diversity of intestinal microbiota, with marked increases in the relative abundance of *Bacteroidetes* [16]. On the other hand, few data in humans have been published regarding the effect of exercise interventions on the intestinal microbiota. In a recent report by Allen et al. a six-week aerobic exercise training altered the intestinal microbiota differently, depending on body weight status [38]. The present study could further elaborate these results by demonstrating that a 12-week aerobic exercise program that consists of brisk walking—in contrast to training of trunk muscles—increased the relative abundance of intestinal *Bacteroides*. This suggests that aerobic exercise may beneficially modify the intestinal microbiota in healthy elderly women. In previous studies, the maintenance of an optimal intestinal environment has been shown to contribute to the prevention of various types of diseases [1,2,3,4,5,6,7]. The results of our study suggest a practical approach, i.e., aerobic exercise, as a strategy to attain the optimization of the intestinal microbiota in humans.

The 12-week aerobic exercise training increased the relative abundance of intestinal *Bacteroides*, in parallel with an improvement in cardiorespiratory fitness. Interestingly, a cross-sectional study has shown that cardiorespiratory fitness is associated with a larger proportion of *Bacteroides* in the intestinal microbiota of premenopausal women [18]. The results from our interventional study are consistent with—and augment—the significance of this observation. We also demonstrated that increases in the relative abundance of *Bacteroides* in the large intestine were greater in the subjects who improved the daily time spent in brisk walking at an intensity of ≥ 3 METs by more than 20 min. On the basis of the findings of the Nakanojo Study, brisk walking at an intensity of > 3 METs for 20 min. or more on most days are recommended for the elderly to reduce the risk of lifestyle-related diseases [39]. In particular, such exercise levels reduce the incidence of osteoporosis [40], metabolic syndrome, hypertension, and hyperglycemia [41]. Based on these considerations, we initially set the target volume of brisk walking for the AE group in our study at 20 min. per day. Our results demonstrate that this regimen also effectively modifies and optimizes the composition of the intestinal microbiota.

By contrast, trunk muscle training did not change the composition of the intestinal microbiota in subjects within our TM group, although it did improve the cardiorespiratory fitness. An improvement in cardiorespiratory fitness in the TM group may have resulted from the strengthening of the respiratory muscles by the trunk muscle training [42]. It may have also occurred because the subjects commuted to the sports center once weekly during the study period. However, the improvement in the cardiorespiratory fitness in the TM group did not coincide with a changed composition of the intestinal microbiota. Furthermore, in the present study, the increase in the time spent in brisk walking was positively correlated with the increase in the relative abundance of intestinal *Bacteroides*. To put these result into perspective, cardiorespiratory fitness may need to be improved by aerobic exercise, such as brisk walking, when the goal is to modify the intestinal microbiota.

There are some candidate mechanisms by which aerobic exercise might increase intestinal *Bacteroides*. Changes in the colonic transit time result in changes in pH within the colonic lumen that may be key in affecting the composition of the intestinal microbiota. Prolonged colonic transit time is known to limit the diversity of intestinal microbiota [43], and this coincides with a greater rise in pH during transit from the proximal to the distal colon [44]. Aerobic exercises, such as jogging and cycling at a moderate intensity, decrease intestinal transit time in healthy people [45] as well as middle-aged patients with chronic constipation [46], probably via increases in the visceral blood flow, increased release of gastrointestinal hormones, mechanical stimulation, and strengthening of the abdominal muscles [46]. Furthermore, aerobic exercise increases the fecal concentrations of the short-chain fatty acids (SCFA) [47], which slightly lowers the colonic-luminal pH [48]. *Bacteroides* species prefer mildly acidic conditions (pH 6.7) for their survival in the colonic lumen, whereas they grow poorly at pH 5.5 [49]. This may explain why aerobic exercise increases intestinal *Bacteroides*, although, of course, a more detailed analysis of the underlying factors remains necessary. 

*Bacteroides* species are opportunistic bacteria: whether they positively or negatively affect host health depends on the characteristics of their intestinal environment. *Bacteroides* spp. play a role in protecting against inflammatory bowel disease [50,51], whereas they may increase infants’ susceptibility to chronic allergic disease, such as early-onset atopic eczema [52]. Thus, future studies will need to detail the clinical consequences of aerobic-exercise-induced increases in *Bacteroides*. Nonetheless, it is widely accepted that lower levels of *Bacteroides* are associated with the higher prevalence of obesity and metabolic syndrome and that *Bacteroides* species may help in suppressing metabolic dysfunction [53,54], although we unfortunately could not evaluate waist circumference as a surrogate index of visceral fat accumulation in the present study. However, in the present study, the increase in intestinal *Bacteroides* in the AE group did not decrease insulin resistance, as assessed by HOMA-IR. This may be because the sedentary but healthy subjects that were included in the present study presented with good insulin sensitivity at baseline, making further improvements in insulin sensitivity following exercise difficult to attain. Further studies should clarify whether aerobic exercise might improve insulin sensitivity through the increase of intestinal *Bacteroides* in obese and/or insulin-resistant subjects.

It is widely accepted that *Bifidobacteria* and *Lactobacillales* contribute to intestinal health, preventing diarrhea and various infectious, allergic, and inflammatory conditions [55]. The relative abundances of these bacteria are decreased in elderly people [8], which may at least partly result in intestinal barrier dysfunction in the population [56]. In addition to some factors, such as probiotics and dietary fiber [56,57], vigorous exercise also has the potential for increasing these bacteria based on rodent studies [14]. However, in the present study, the relative abundance of *Bifidobacterium*, the only *Bifidobacteria* that can be identified by our T-RFLP analysis, as well as *Lactobacillales* remained unchanged in both groups. This may be because the quantity (time and intensity) of our brisk walking was not enough to increase these bacteria. Exploring an exercise prescription that can increase these bacteria will benefit intestinal health in elderly people.

A few limitations of the present study should be noted. First, our non-randomized study design with a relatively small subject sample size may have been insufficiently powered to detect differences in efficacy between the two exercise programs to affect the clinical outcomes, such as trunk muscle strength, cardiorespiratory fitness, defecation pattern, and the composition of the intestinal microbiota. Second, we confirmed that the participants had no substantial exercise habits before exercise intervention. However, exercise-induced energy expenditure at the baseline was greater in the AE than in the TM group. The reason for this may have been that the subjects who opted for the AE training were more aware of the health benefits of walking, which may have resulted in a superior effect of brisk walking to trunk muscle training on increasing intestinal *Bacteroides*. Finally, we classified fecal intestinal microbiota into only 10 major groups, which were present in the fecal samples of all the subjects. Therefore, it was impossivly to evaluate the effects of the exercise intervention on the diversity of the intestinal microbiota. A greater diversity of the intestinal microbiota is generally considered to provide various health benefits. It might have been possible to detect an increase in the diversity of the intestinal microbiota following the exercise intervention if the adopted microbiotic classification scheme had included several hundred subdivisions.

## 5. Conclusions

Aerobic exercise training targeting an increase of the time spent in brisk walking may have a potential for increasing intestinal *Bacteroides,* while also improving cardiorespiratory fitness in healthy elderly women. Exercise intervention may provide a practical means of acquiring a more optimal composition of intestinal microbiota. Further studies are needed to clarify the mechanism by which exercise exerts it effect on the composition of the intestinal microbiota.

## Figures and Tables

**Figure 1 nutrients-11-00868-f001:**
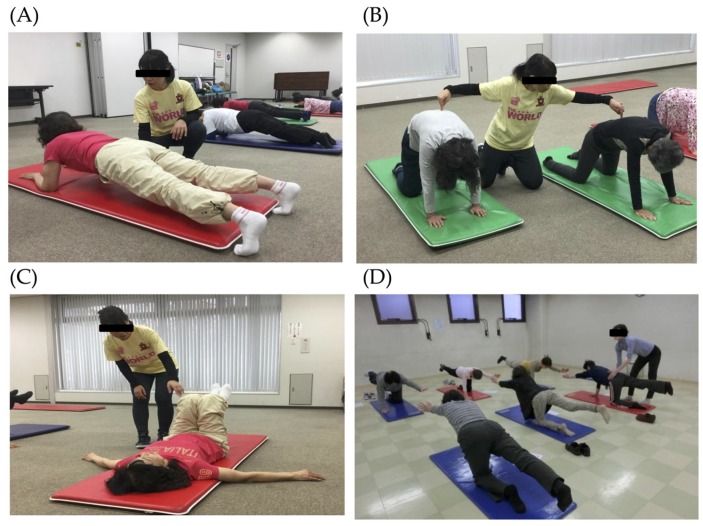
Exercises during trunk muscle training. (**A**) Arching–swaying while standing on all fours, (**B**) plank, (**C**) lying pelvic rotation, and (**D**) diagonal lifting while standing on all fours.

**Figure 2 nutrients-11-00868-f002:**
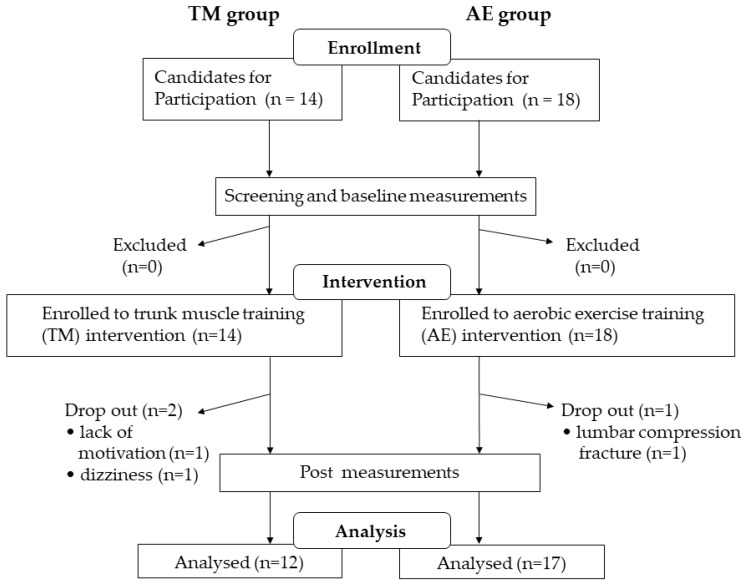
Flowchart of the screening, enrollment, intervention, and data analysis of the study. Abbreviations: TM, trunk muscle training; AE, aerobic exercise training.

**Figure 3 nutrients-11-00868-f003:**
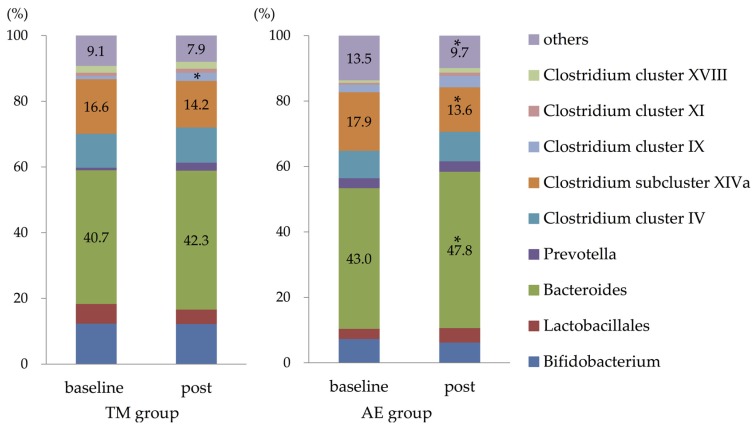
Changes in the composition of the intestinal microbiota following the intervention. The relative abundance of intestinal *Bacteroides* was significantly increased, and that of the *Clostridium* subcluster XIVa was decreased only in the AE group (by the Wilcoxon Signed-rank test). The relative abundance of *Clostridium* cluster IX was significantly increased only in the TM group. *: *p* < 0.05 compared with baseline. Abbreviations: TM, trunk muscle training; AE, aerobic exercise training.

**Figure 4 nutrients-11-00868-f004:**
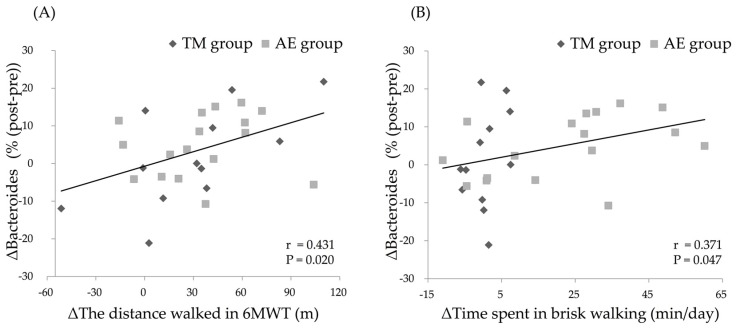
Relationship between changes in the distance during the 6MWT (6MWD) (**A**), changes in the time spent in brisk walking (**B**), and changes in the relative abundance of intestinal *Bacteroides* by the intervention. Improvements in 6MWD and time spent in brisk walking were positively correlated with increases in the relative abundance of intestinal *Bacteroides* in all subjects. Abbreviations: TM, trunk muscle training; AE, aerobic exercise training; 6MWT, 6-min. walk test; 6MWD, distance in the 6MWT.

**Figure 5 nutrients-11-00868-f005:**
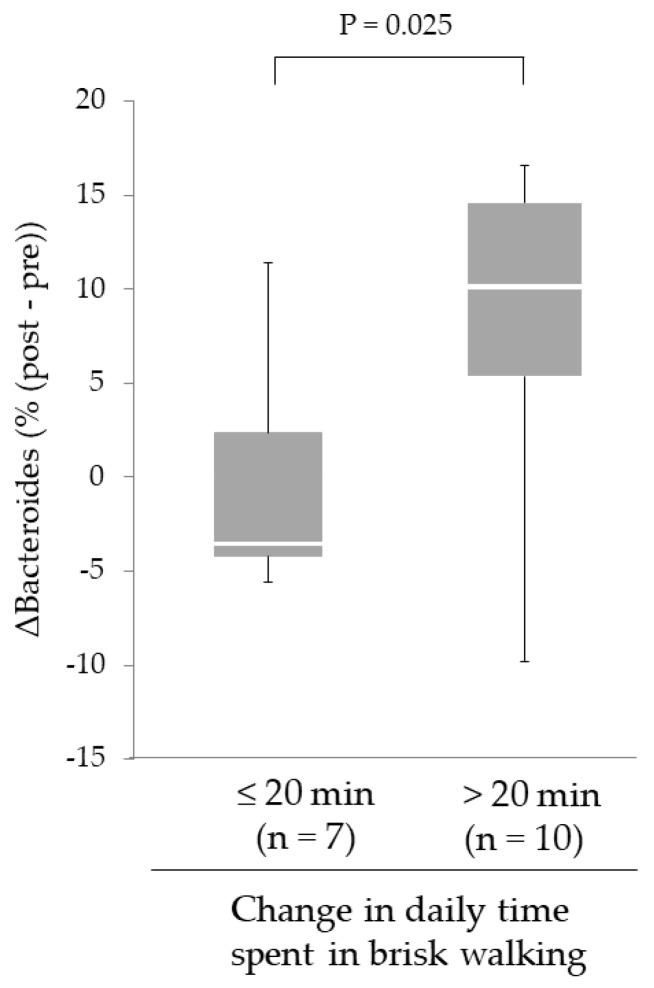
Effect of increased daily physical activity on changes in the relative abundance of intestinal *Bacteroides* following the intervention in the AE group. Increases in intestinal *Bacteroides* in subjects who increased the daily time spent in brisk walking for 20 min. or more were greater than in those who did not (by the Mann–Whitney *U*-test). Horizontal bars indicate the minimum values, the 25th, 50th, 75th percentile levels, and the maximum values. Abbreviations: AE, aerobic exercise training.

**Table 1 nutrients-11-00868-t001:** Clinical characteristics of the subjects.

		Total	TM Group	AE Group
*n*		29	12	17
Age	(years)	70 (66–75)	70 (66–77)	70 (66–75)
BW	(kg)	51.8 (47.8–56.5)	49.8 (48.3–56.8)	52.0 (46.9–56.0)
BMI	(kg/m^2^)	21.4 (18.8–23.1)	20.6 (18.7–24.0)	21.7 (18.9–23.1)
Body fat	(%)	29.0 (23.6–32.7)	26.6 (22.9–32.2)	30.6 (25.1–33.0)
SBP	(mmHg)	141 (120–152)	129 (114–151)	142 (124–154)
DBP	(mmHg)	82 (74–92)	81 (74–86)	85 (74–93)
Present illness	*n* (%)			
No		17 (58.6)	9 (75.0)	8 (47.1)
Yes		12 (41.4)	3 (25.0)	9 (52.9)
Past history	*n* (%)			
No		15 (51.7)	7 (58.3)	8 (47.1)
Yes		14 (48.3)	5 (41.7)	9 (52.9)
Medication	*n* (%)			
No		19 (65.5)	10 (83.3)	9 (52.9)
Yes		10 (34.5)	2 (16.7)	8 (47.1)

Data are presented as median (interquartile range) for age, BW, body fat, SBP, and DBP, and as *n* (%) for present illness, past history, and medication. Abbreviations: TM, trunk muscle training; AE, aerobic exercise training; BW, body weight; BMI, body mass index; SBP, systolic blood pressure; DBP, diastolic blood pressure.

**Table 2 nutrients-11-00868-t002:** Changes in the parameters following the intervention.

		TM Group (*n* = 12)		AE Group (*n* = 17)	
		Baseline	Post	Baseline	Post
BMI	(kg/m^2^)	20.6 (18.7–24.0)	20.8 (18.8–23.8)	21.7 (18.9–23.1)	21.3 (18.8–23.5)
Body fat	(%)	26.6 (22.9–32.2)	27.4 (23.7–31.9)	30.6 (25.1–33.0)	28.6 (25.1–33.75)
Leg muscle mass	(kg)	8.08 (7.06–8.29)	7.82 (6.80–8.16)	7.29 (7.03–8.08)	7.44 (7.12–8.25)
K-W test score	(/40)	15.5 (8.5–24.8)	27.5 (22.0–31.8) *	13.0 (9.0–16.5)	21.0 (15.5–29.0) *
Quad. muscle strength	(kg)	22.7 (20.1–29.2)	23.5 (22.1–30.8)	26.2 (19.9–32.5)	24.8 (20.6–29.2)
MSL	(cm)	111.6 (107.6–123.2)	111.5 (107.0–125.5)	112.9 (108.9–120.0)	113.1 (104.3–119.5)
TUG	(sec)	6.19 (5.60–6.77)	5.80 (5.40–6.50)	6.14 (5.50–6.80)	5.87 (5.59–6.42)
Single-leg standing	(sec)	28.6 (12.3–120.0)	70.9 (32.3–120.0)	98.5 (39.9–120.0)	120.0 (79.0–120.0)
6MWD	(m)	540.8 (521.0–570.0)	567.5 (538.0–627.6) *	550.0 (510.9–579.7)	582.7 (541.0–618.7) *
Number of steps	(steps/day)	6348 (5256–7267)	6438 (4443–8073)	7869 (6456–10246)	10297 (7396–14117) *
Time spent in brisk walking	(min/day)	10 (2–15)	9 (2–17)	16 (8–30)	45 (16–52) *
Total EE	(kcal/day)	1561.0 (1418.3–1672.8)	1561.5 (1406.3–1613.3) *	1598.0 (1478.0–1724.0)	1633.0 (1469.5–1844.0) *
Exercise-induced EE	(kcal/day)	125.5 (99.5–140.0)	125.5 (85.5–154.0)	161.0 (118.5–211.5)	228.0 (153.5–318.0) *
FPG	(mmol/L)	5.9 (5.5–7.0)	5.7 (5.3–6.8)	5.8 (5.2–6.1)	5.3 (5.1–6.3)
TG	(mmol/L)	1.08 (0.87–1.27)	1.07 (0.91–1.54)	0.89 (0.75–1.17)	1.06 (0.91–1.53)
LDL-C	(mmol/L)	3.45 (3.23–3.77)	3.40 (2.95–4.25)	3.72 (3.25–4.19)	3.72 (3.21–4.24)
HDL-C	(mmol/L)	1.60 (1.27–2.26)	1.66 (1.29–2.43)	1.73 (1.42–2.03)	1.68 (1.44–2.06)
Insulin	(pmol/L)	29.8 (21.7–33.7)	32.3 (25.8–60.4)	38.0 (26.2–54.5)	40.2 (25.1–59.6)
HOMA-IR		1.10 (0.74–1.45)	1.14 (0.86–2.55)	1.36 (0.84–2.05)	1.31 (0.80–2.32)

All values are presented as median (interquartile range). Changes in clinical parameters following intervention in each group were examined by the Wilcoxon Signed-rank test. *: *p* < 0.05 compared with baseline. Abbreviations: TM, trunk muscle training; AE, aerobic exercise training; BMI, body mass index; K-W test score, Kraus–Weber test score; Quad. muscle strength, Quadriceps muscle strength; MSL, maximal step length; TUG, Timed Up & Go; 6MWD, distance in the 6-min. walk test; EE, energy expenditure; FPG, fasting plasma glucose; TG, triglyceride; LDL-C, low-density lipoprotein cholesterol; HDL-C, high-density lipoprotein cholesterol; HOMA-IR, homeostatic model assessment of insulin resistance.

**Table 3 nutrients-11-00868-t003:** Changes in nutrient intake and defecation pattern following the intervention.

		TM Group (*n* = 12)		AE Group (*n* = 17)	
		Baseline	Post	Baseline	Post
Nutrient intake					
Total energy	(kcal/day)	1863 (1827–1908)	1878 (1839–1942)	1874 (1795–1956)	1828 (1796–1942)
Carbohydrates	(g/day)	244.8 (237.6–252.7)	248.0 (243.0–255.3)	246.7 (240.8–258.1)	243.4 (238.9–255.2)
Protein	(g/day)	76.5 (74.2–83.1)	76.8 (74.4–84.2)	75.6 (71.6–82.9)	75.3 (71.8–82.1)
Lipid	(g/day)	59.2 (57.8–60.5)	59.9 (59.1–64.5)	58.9 (56.2–64.0)	58.5 (55.8–63.9)
Saturated fat	(g/day)	17.1 (16.7–17.7)	17.7 (16.7–20.0)	17.7 (16.1–19.1)	16.9 (16.0–19.1)
Fiber	(g/day)	17.6 (17.1–17.9)	18.2 (17.3–18.8)	17.6 (17.2–18.5)	17.7 (17.0–18.2)
Defecation pattern					
CAS-J	(/16)	3.50 (2.25–5.75)	3.50 (2.00–5.75)	2.00 (1.00–4.50)	2.00 (0.00–3.00) *
Abdomen appears distended or swollen	(/2)	0.0 (0.0–1.0)	0.0 (0.0–1.0)	0.0 (0.0–1.0)	0.0 (0.0–0.0)
Amount of flatus	(/2)	1.0 (0.0–1.0)	1.0 (0.0–2.0)	0.0 (0.0–1.0)	0.0 (0.0–1.0)
Frequency of defecation	(/2)	0.0 (0.0–1.0)	0.0 (0.0–1.0)	0.0 (0.0–1.0)	0.0 (0.0–0.5)
Rectum appears to be filled with feces	(/2)	1.0 (0.0–1.0)	0.0 (0.0–0.8) *	0.0 (0.0–1.0)	0.0 (0.0–0.0)
Pain of the anus during defecation	(/2)	0.0 (0.0–1.0)	0.0 (0.0–0.8)	0.0 (0.0–0.5)	0.0 (0.0–0.0)
Amount of feces	(/2)	0.0 (0.0–0.8)	0.0 (0.0–1.0)	0.0 (0.0–0.0)	0.0 (0.0–0.0)
Ease of defecation	(/2)	0.5 (0.0–1.0)	0.0 (0.0–1.0)	0.0 (0.0–1.0)	0.0 (0.0–1.0) *
Diarrhea or watery stools	(/2)	0.0 (0.0–1.0)	0.0 (0.0–0.0)	0.0 (0.0–1.0)	0.0 (0.0–0.0)

All values are presented as median (interquartile range). Changes in clinical parameters following intervention in each group were examined by the Wilcoxon Signed-rank test. *: *p* < 0.05 compared with baseline. Abbreviations: TM, trunk muscle training; AE, aerobic exercise training; CAS-J, Japanese version of the Constipation Assessment Scale.

**Table 4 nutrients-11-00868-t004:** Correlation coefficients in simple regression analysis between clinical factors and the changes in the relative abundance of intestinal *Bacteroides* following the exercise intervention in all subjects.

Related Factors	Correlation Coefficient	*p* Value
Age	−0.343	0.068
Pre-*Bacteroides*	−0.519	0.004 *
ΔK-W test score	0.327	0.083
Δ6MWD	0.431	0.020 *
ΔNumber of steps	0.210	0.275
ΔTime spent in brisk walking	0.371	0.047 *
ΔTotal EE	0.216	0.261
ΔExercise-induced EE	0.250	0.191
ΔCAS-J	0.071	0.715

The relationships between the parameters and changes in the relative abundance of specific types of intestinal microbiota were examined by Spearman’s rank correlation coefficient test. *: *p* < 0.05. Abbreviations: Pre-*Bacteroides*, the relative abundance of the intestinal *Bacteroides* before the intervention; K-W test score, Kraus–Weber test score; 6MWD, distance in the 6-min. walk test; EE, energy expenditure; CAS-J, Japanese version of the Constipation Assessment Scale.

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
