# Peer review of "Aerobic Exercise Training with Brisk Walking Increases Intestinal Bacteroides in Healthy Elderly Women"

_nutrients, 2019, doi:10.3390/nu11040868_

Reviewer 1 Report

The manuscript examines the impact of two types of exercise on fitness and faecal bacteria. 

There are some recent studies that were not discussed in the introduction.  However, this is a key area of interest especially as exercise can reduce risk of colon cancer.

There are several methodological weaknesses.

The subjects were not randomised to the two types of exercise so it would be better if the two types of exercise were presented as two different studies and not compared directly.   

It is not clear how the energy measure was calculated nr what type of accelerometer was used. 

The numbers in each trial are small and not powered to detect differences in several measurements especially to compare between group, and for changes in intake.

The bacteriology methods are not clear.  The faecal samples were preserved in an unknown preservative, without comment on the relative loss of species during storage nor was the storage at 4C for 7 days validated as being suitable for storage of samples.  The DNA methods were not of the highest standard allowing only identification at higher group level which limits the interpretation.   The levels of bifidobacteria are of particular interest in the elderly as they decrease with age and may be more important than bacteroides.  This was not discussed.  There is over interpretation of the results given such small numbers of subjects.

Author Response

We greatly appreciate Reviewer 1’s kind comments and suggestions. We revised our manuscript according to the comments, and would like to address the points as below.

 1.        According to Reviewer 1’s comment, we cited recent reports especially about the effect of exercise on incidence of colon cancer, and discussed about them in the Introduction section of the revised manuscript as follows:

 Line 58-62

 2.        As Reviewer 1 pointed out, a direct comparison between the groups may have been inappropriate due to our nonrandomized design with the small number of samples. Therefore, we decided only to perform the comparison between before and after the intervention within each group using a nonparametric test in the revised manuscript. Along with this changes, we revised 2.11. Statistical analyses and the Results section. Table2-4 and figures3-4 were also replaced by new ones.

Careful interpretation of our results is also needed due to due to our nonrandomized design with the small number of samples. Therefore, we toned down the descriptions in the Abstract and the Conclusions sections as follows:

 Line 38

Line 484-485

  3.        Regarding the energy measure, we used a three-axis accelerometer without a barometric nor a gyro sensor to quantify physical activity including metabolic equivalents (METs). This device also automatically calculates energy expenditure (EE) from METs based on a widely-accepted formula (EE (kcal) = 1.05 × METs × time (h) × body weight (kg), American College of Sports Medicine position stand. Med Sci Sports Exerc, 2011). The estimated EE was highly correlated with that evaluated by indirect calorimetry using a gas analyzer (Takeshima N, et al., 2017. Japanese). This accelerometer has also been adequately validated (Sakaki S, et al. 2016 (ref 22), Moriyama N, et al. 2017 (ref 23)). According to Reviewer 1’s comment, we revised 2.7. Daily physical activity level in the Materials and Methods section as follows:

 Line 196-198

4.        Regarding the bacteriology methods, guanidine thiocyanate was used as a preservative. This solution is common as a simple and safe preservative for fecal storage at room for DNA analysis. It is reported that the relative abundance of Bifidobacterium appeared to decrease when samples are stored for as long as 2 to 4 weeks (Hosomi K, et al. 2017 (ref 25)). Therefore, our storage, in which fecal samples were kept refrigerated at 4°C until transfer to the laboratory up to 7 days, is thought minimally to affect the composition of the intestinal microbiota. According to Reviewer 1’s comment, we revised the 2.4. Analysis of intestinal microbiota paragraph in the Materials and Methods section as follows:

 Line 131-134

     As Reviewer 1 pointed out, the identification sensitivity of our T-RFLP analysis might be inferior to that of the highest standard, although our analysis is one of the most well-established and reliable 16S ribosomal RNA-based methods. To address Reviewer 1’s comment, we added some descriptions as a limitation on the Discussion section as follows:

 Line 136-138

Line 462-463

 5.        As Reviewer 1 pointed out, Bifidobacteria is one of the health-promoting bacteria which is known to decrease with aging. In this study, the relative abundance of Bifidobacterium, the only Bifidobacteria which can be identified by our T-RFLP analysis, remained unchanged throughout the intervention in both groups, although rodent studies have shown that a vigorous exercise training increased the relative abundance of Bifidobacteria. This may be because the quantity (time and intensity) of our brisk walking was insufficient to increase Bifidobacteria. We added this issue to the Discussion section as follows:

 Line 457-466

Reviewer 2 Report

The manuscript entitled “Aerobic exercise training with brisk walking increases intestinal Bacteroides in healthy elderly women” submitted by Morita et al., ascertains evidence about the benefits of the aerobic exercise training related to the gut microbiota. Aerobic exercise increased the intestinal Bacteroides while also improved cardiorespiratory fitness in healthy ederly women. Nevertheless, there are some minor points that need to be addressed.

1-Introduction section: please, include more literature related to the well-known evidence between microbiota and exercise (The Effects of Dietary Pattern during Intensified Training on Stool Microbiota of Elite Race Walkers. Murtaza N, Burke LM, Vlahovich N, Charlesson B, O' Neill H, Ross ML, Campbell KL, Krause L, Morrison M. Nutrients. 2019 Jan 24;11(2). pii: E261. doi: 10.3390/nu11020261; Different host factors are associated with patterns in bacterial and fungal gut microbiota in Slovenian healthy cohort. Mahnic A, Rupnik M. PLoS One. 2018 Dec 20;13(12):e0209209. doi: 10.1371/journal.pone.0209209)

2-Methods: Thirty-two healthy sedentary women aged 65 years and over were recruited from Osaka City (Japan) and they were randomly allocated in two groups: TM group and AE group. What about a control group? I think that this point should be taking account to compare the baseline conditions regarding microbiota composition.

3-Regarding anthropometrical measurements: The waist/hip ratio should be included.

4-Surprinsingly, as it has been described in rodents, the levels of Bifidobacterium and Lactobacillus did not change in the AE group. Maybe, this is due to the time spent in brisk walking (20 min) and the major benefit of this is stronger at the osteoporosis, metabolic syndrome, hypertension, and hyperglycemia. Maybe, 20 min is not enough to change all the gut microbiota. This could be discussed better in the Discussion section.

 Author Response

We greatly appreciate Reviewer 2’s time and efforts to review our manuscript. We revised our manuscript according to the comments, and would like to address the points as below.

 1.        According to the suggestion by the Reviewer 2, we refered to 2 literatures in the Introduction section in the revised manuscript as follows:

Line 68-71

 2.        Regarding a control group, we did not arrange the control because we aimed to examine whether the exercise modality influence the effect of training on the composition of intestinal microbiota in sedentary elderly women. However, a direct comparison between the groups may have been inappropriate due to our nonrandomized design with the small number of samples. Therefore, we decided only to perform the comparison between before and after the intervention within each group using a nonparametric test in the revised manuscript. Along with this changes, we revised 2.11. Statistical analyses and the Results section. Table2-4 and figures3-4 were also replaced by new ones.

 3.        Lower levels of Bacteroides are associated with higher prevalence of metabolic syndrome. Unfortunately we did not measure the waist/hip ratio, however, Reviewer 2 pointed out, evaluation of visceral fat accumulation could augment our discussion. Regarding this issue, we added some descriptions in the Discussion section as follows:

   Line 449-450

   4.        Both Bifidobacterium and Lactobacillus are the health-promoting bacteria which is known to decrease with aging. In this study, the relative abundance of Bifidobacterium, the only Bifidobacteria which can be identified by our T-RFLP analysis, as well as Lactobacillales remained unchanged throughout the intervention in both groups, although rodent studies have shown that a vigorous exercise training increased the relative abundance of Bifidobacteria. As Reviewer 2 pointed out, this may be because the quantity (time and intensity) of our brisk walking was insufficient to increase Bifidobacteria. We added this issue to the Discussion section as follows:

 Line 457-466

Round  2

Reviewer 1 Report

The authors have improved the manuscript and the analysis of their study.  In the introduction they have improved their discussion of the association of exercise or fitness with the microbiota and emphasized that no studies have clearly shown impact of change in exercise on the microbiota.  This could be strengthened further by stating that when comparing for example rugby players with sedentary individuals there are many potential confounding factors including diet which may have more impact on the microbiome than exercise.

The statistical analysis is now performed using non-parametric methods.  The presentation of the data in the text tables and figures should reflect this and not use means and SD but rather median and interquartile ranges. 

Figure 2 seems to have two versions overlaid but should be modified to emphasis that  these are two separate studies by separating the two flow charts completely.

Table 1 needs to be modified to remove comparison of the two groups of subjects.

All other tables need to be modified to remove comparison between groups and reflect the new non parametric analysis.  It would be useful to state the stats tests used in the footnotes to be clear.

Author Response

We thank Reviewer 1 again for time and effort to re-review our manuscript. Here, we would like to address point-by-point to Reviewer 1’s comment.

 1. As Reviewer 1 pointed out, most of previous studies could not differentiate the effect of exercise on intestinal microbiota from that of eating habits. According to Reviewer 1’s suggestion, we revised the Introduction section as follows:

 Line 70 – 73

  2. According to Reviewer 1’s comments, we used medians and interquartile ranges instead of means and SDs in the revised manuscript. Along with this changes, we revised 2.11. Statistical analyses and the Results section. Table1-3 and figures5 were also replaced by new ones.

   Line 238, 256-257, 366-367

  3. As Reviewer 1 pointed out, our study design was non-randomized and we have to separate the flow chart to two independent ones. Therefore, we replaced Figure 2 to new one.

4. According to Reviewer 1’s comment, we removed all inter-group comparisons in the revised manuscript. We now used Wilcoxon’s test for within group comparison and presented the data as medians and interquartile ranges. Along with this changes, we revised 2.11. Statistical analyses and the Results section. We also replaced Figure 2 and Figure 5 as well as Table 1-3. Our revisions are follows:

     Line 238-240

    Line 256-257

    Line 275, 278-279, 281, 284-285

    Line 292

    Line 296-297

    Line 302-303, 306, 308, 315

    Line 346-347

    Line 359-360, 366-367

    Line 465

Nutrients EISSN 2072-6643 Published by MDPI AG, Basel, Switzerland RSS E-Mail Table of Contents Alert
Back to Top